# Physiological Responses of Chickpea Genotypes to Cold and Heat Stress in Flowering Stage

**Mareen Zeitelhofer \*, Rong Zhou \*** and **Carl-Otto Ottosen**

Department of Food Science, Aarhus University, 8200 Aarhus, Denmark
\* Correspondence: m.zeitelhofer@hotmail.com (M.Z.); rong.zhou@food.au.dk (R.Z.)

**Abstract:** Due to climate change, more temperature extremes are expected in the future, potentially endangering agricultural production. Chickpea (*Cicer arietinum* L.) is an important cool-season food legume grown worldwide; however, cold and heat episodes are major threats in chickpea production that cause considerable yield losses especially at the flowering stage. The aim of this study was to evaluate the physiological performance of contrasting chickpea genotypes during the flowering phase under cold and heat. Four chickpea genotypes (Desi, Eldorado, Acc#2 and Acc#7) with different temperature susceptibilities were treated for 3 days under cold (9/4 °C) and heat (38/33 °C). The results showed that cold stress reduced the maximum quantum efficiency of photosystem II ($F_v/F_m$) by 5%, net photosynthetic rate ($P_N$) by 74%, and chlorophyll a+b content by 31% on average in all tested genotypes. Up to a 9-fold increase in the amount of starch was found in the leaves of plants under cold stress, indicating that carbohydrates strongly accumulated in chickpeas under cold stress. This helps to maintain the vegetative and generative organs and enable fast recovery. Under heat stress, chickpeas maintained $F_v/F_m$ and $P_N$, although chlorophyll a+b content decreased by 39% on average. Carbohydrates did not accumulate under heat in chickpeas; thereby, a reduction in biomass and reproductive organs took place. Genetic variation in response to cold and heat stress was detected among the tested flowering chickpea genotypes. Desi and Acc#2 were cold-sensitive candidates, and Eldorado was a cold-tolerant candidate, whereas Acc#7 and Acc#2 were heat-sensitive candidates, while Desi and Eldorado were heat-tolerant candidates. This study provides important knowledge on the physiological response of flowering chickpeas under cold and heat stress. This will benefit the identification of stress-tolerant chickpea genotypes to ensure high yields in the future climate.

**Keywords:** chickpea; cold stress; heat stress; flowering stage; plant physiology

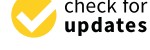



## 1. Introduction

Changes in the frequency, intensity and duration of weather extremes are expected due to climate change [1]. Temperature is one of the most important factors affecting plant growth [2]. Extreme variations in temperature, both low and high temperature, can have a serious impact on plant development, whereas the reproductive phase is more sensitive than the vegetative one, often leading to reduction in yield [3,4]. Cold stress induced by low temperatures affects different aspects of the photosynthesis apparatus, making photosystem I (PSI) and photosystem II (PSII) strongly temperature-dependent, possibly leading to photodamage [3]. Cold stress caused increased membrane viscosity, retarded metabolism, delayed energy dissipation and radical formation as well as oxidation stress in plant tissues [5]. Photosynthesis in plants is limited by heat stress due to high temperatures, as it is the most thermosensitive part of plant function, whereas PSII is more sensitive than PSI [6]. Heat stress decreases the amounts of photosynthetic pigments but increases the accumulation of soluble carbohydrates in the leaves, ensuring stress tolerance through osmotic adjustment [7]. From a climate change perspective, it is crucial to detect varieties that can deal with the temperature extremes at different stages of development to ensure food security in the future [4,8].

Chickpea (*Cicer arietinum* L.) is an important leguminous grain crop for more than 50 countries [9]. It is grown on 14.56 million ha worldwide, with 11.5 million tons harvested per year and most of the production centered in India [10]. Climate change is a major challenge in chickpea production nowadays [11]. Chickpea is classified as a cool-season food legume and is limited by major abiotic stresses, including cold and heat stress [9,12]. The ability of legumes to adapt to the predicted extremes of temperatures is largely unknown [13]. Daily maximum temperatures above 25 °C were seen as the threshold level for heat stress in cool-season food legumes; however, both high- and low-temperature stress can affect their productivity [14]. Chickpea's sensitivity to supraoptimal temperatures, especially during its reproductive stage, can lead to drastic yield losses, which currently limit cultivation in the temperate zone [15]. Both high and low temperatures (>30 °C or <15 °C) have been seen as the major constraints for chickpea production, leading to considerably reduced chickpea yields [16,17]. However, limited research has been conducted to detect cold- and heat-tolerant chickpea genotypes. Large genetic variation was found among chickpea genotypes subjected to heat stress with respect to phenology, growth, yield components and grain yield [18]. The genetic variation in chickpeas at pod set at low temperatures was identified both under field and controlled conditions; however, the morpho-physiological basis for the variation was unclear [19]. From these perspectives, it is crucial to evaluate the physiology of flowering chickpeas under cold and heat stress and to identify candidates that can deal with these extremes to maintain and increase chickpea yields in the future.

This study focused on the physiological response of a panel of four chickpea genotypes selected in previous studies at flower stage to cold and heat stress [20,21]. Our aim was to evaluate their physiological performance under cold and heat stress and to identify stress-tolerant vs. stress-sensitive flowering chickpea candidates. We hypothesized that (a) cold and heat stress have a differential negative impact on the physiology of flowering chickpeas; (b) the physiological response of chickpeas to temperature stress depends on the genotype. This study will provide new insights for selecting resilient chickpeas that can deal with predicted temperature extremes in the future.

## 2. Materials and Methods

Based on previous screening experiments, the following four genotypes were selected: Eldorado (Iniav, Lisbon, Portugal; Kabuli type, late-flowering), Desi (Arche Noah, Schiltern, Austria; Desi type, early-flowering), Acc#2 and Acc#7 (ICRISAT, Patancheruvu, India; Desi type, early-flowering). The seeds were sown in plastic pots (11/9 cm, diameter/height) with commercial peat substrate (Pindstrup 2; Pindstrup Mosebrug A/S, Ryomgaard, Denmark).

Based on the screening experiments from Makonya et al. (2019), the following parameters were set: The climate parameters of the greenhouse were set to 23/16 °C (day/night), ambient $CO_2$ concentration (405 ppm), and 50 ± 10% relative humidity (RH). Supplementary light was provided with broadband white LED lamps (FL300 sunlight, Senmatic, Søndersø, Denmark), which were turned on automatically when the natural light intensity was below 150 µmol m$^{-2}$ s$^{-1}$. The light level was 304 ± 19 µmol m$^{-2}$ s$^{-1}$ during the day period in the greenhouse. The plants were irrigated once a day with the following nutrient solution: pH = 6, EC = 2.18, $NH_4$ = 10.9%, N = 191 ppm, P = 35 ppm, K = 275 ppm, Mg = 40 ppm, Ca = 140 ppm.

Due to the differences in growth and development speed, the chickpea genotypes were treated at different plant ages to ensure that all of the genotypes were at a similar developmental stage (1/3 flowering stage to 100% flowering stage) before the stress treatment. The 31-day- old Acc#7, 38-day-old Desi, 41-day-old Acc#2 and 48-day-old Eldorado were moved to the climate chambers (MB teknik, Brøndby, Denmark) for treatments. The plants were treated at (1) 25/20 °C (day/night) (control, CON); (2) 9/4 °C (day/night) (cold-stress treatment, CS) and (3) 38/33 °C (day/night) (heat-stress treatment, HS). The treatments started at 12:00 o'clock and lasted for 3 days (72 h). The light level was set to 300 µmol m$^{-2}$ s$^{-1}$ using broadband white LED lamps (FL300 sunlight, Senmatic, Søndersø,

Denmark) within all treatments, and RH was set to 62%/53% (day/night) to maintain equal VPD at CON and CS and to 80%/70% (day/night) at HS. The actual temperature, RH and light intensity in the climate chambers are shown in Supplementary Table S1. Eight uniformly sized plants were randomly selected per genotype and treatment. The plants were watered twice per day in the control and cold-stress treatment and three times per day in the heat-stress treatment to avoid water deficit.

Chlorophyll fluorescence

The second fully expanded upper leaflet was used for chlorophyll fluorescence measurements by miniPAM (Heinz Walz, Eifeltrich, Germany). The measurements were performed under control conditions, after 24 h stress application (SD-1), after 48 h stress application (SD-2), after 72 h stress application (SD-3) and after 48 h recovery time under control conditions (REC) between 11:30 and 13:00 o'clock. The leaves were dark-adapted for 30 min with a leaf clip before the measurements of $F_v/F_m$ (maximum quantum efficiency of photosystem II or PSII), with eight replications taken. For each plant, two leaflets were measured, and the results were averaged per plant.

Gas exchange

The second fully expanded top leaf was chosen for gas exchange measurements with three (CS, HS) and eight (CON) replications starting from 69 h of the treatments and lasting for at least 3 h. The parameters $P_N$ (net photosynthetic rate), $C_i$ (intracellular $CO_2$ concentration), $g_s$ (stomatal conductance) and E (transpiration rate) were measured using a portable photosynthesis system (CIRAS-2, PP Systems, Amesbury, MA, USA). The cuvette temperature settings were 25 °C for CON, 9 °C for CS and 38 °C for HS. The light level was provided by LED light unit (PP Systems, Hitchin, UK) mounted on the leaf cuvette and set to 300 μmol m$^{-2}$ s$^{-1}$. The leaves were placed in 1.7 cm$^2$ cuvettes with 400 μmol m$^{-2}$ s$^{-1}$ $CO_2$ concentration and 200 cm$^3$/min airflow rate. Records were taken every 10 s at steady state. In total, $2^1/_2$–5 min recorded measurements were averaged.

Leaf pigments and carbohydrates

The second fully expanded top leaf was harvested for pigment and carbohydrate content measurements after 3 days of treatments. The samples were directly frozen in liquid nitrogen and stored at −80 °C until analysis. The samples were dried for 3 days in a freezer dryer at −54 °C (Gamma 1–20, LMC-1, Ballerup, Denmark).

For leaf pigment determination, dry weight of the samples was taken with four replicates. Extraction was carried out by adding 1.8 mL 96% ethanol to each sample (7–8 mg) until the leaf material turned pale. From each sample, 750 μL of the supernatant were taken and put into a cuvette. All samples were diluted eight times with 96% ethanol. The extraction was analyzed for pigments using a spectrophotometer (UV-VIS Spectrophotometer, Shimadzu, Koyto, Japan) at 470 nm, 648.6 nm, 664.2 nm, and 750 nm absorbances. Chlorophyll a (Chl. a), chlorophyll b (Chl. b), total chlorophyll (Chl. a+b), chlorophyll a:b ratio (Chl. a:b ratio) and carotenoid content were calculated according to Lichtenthaler (1987) [22]. The following equations were used:

$$\text{Chl. a} = 13.36A664.2 - 5.19A648.6$$

$$\text{Chl. b} = 27.43A648.6 - 8.12A664.2$$

$$\text{Chl. a+b} = 5.24A664.2 + 22.24A648.6$$

$$\text{Chl. a:b ratio} = (1000A470 - 2.13\text{Cl. a} - 97.64\text{Cl. b})/209$$

For carbohydrate determination, 5–10 mg DW of finely ground tissue of 5 replicates was weighed out in an Eppendorf vial. Afterwards, the samples were extracted with ethanol four times following these steps:

(1.)  Add 400 μL of the 80% ETOH solution (for 200 mL: 160 mL 96% ethanol + 40 mL HEPES stock solution) and vortex.
(2.)  Shake for 15 min at 80 °C in an Eppendorf Thermo Mixer at 1300 rpm and centrifuge 1–3 min at 12,000 rpm.

(3.) Collect the supernatant in a 2 mL Eppendorf vial, and keep it on ice in the dark.

(4.) Extract the remaining residue again with 400 μL of the 50% ETOH solution (for 50 mL: 25 mL 96% ethanol, 10 mL HEPES stock solution, 15 mL dd $H_2O$).

(5.) Repeat step 2 and collect the supernatant and pool it with the first one.

(6.) Repeat ethanol extraction (heating included) twice with 200 μL of 80% ETOH until the pellet is clear.

The supernatants were filled to 1.5 mL with 80% ETOH and stored at −80 °C for sugar analysis, while the pellets were left to dry at room temperature for starch analysis. The supernatants were diluted and filtered with a 0.45 μm nylon filter. They were analyzed through ion chromatography (Dionex, ICS 3000, town country) for soluble sugars (sucrose, glucose and fructose). The chromatograph was equipped with a pulsed amperometric detector (PAD), with a working gold electrode operating in the integrated amperometric mode.

For the starch analysis, the following steps were followed:

(1.) Add to the pellets in the Eppendorf vials 1000 μL dd $H_2O$ plus one metal ball and put in the ball mill (Resch, 200 mm) for 2 min at 22 Hz.

(2.) Autoclave the Eppendorf vials for 90 min, cool and vortex them afterwards.

(3.) Place 100 μL sample plus 400 μL buffer enzyme solution (needs to be freshly prepared, 400 μL Na-Acetate, 0.184 mg Amyloglucosidase, 0.16 μL α-amylase) into 2 mL Eppendorf tubes.

(4.) Incubate for 16 h at 37 °C and 750 rpm in the Thermo Mixer.

(5.) Centrifuge the samples for 5 min at 13,000 rpm until the solution is clear

(6.) Diluted and filter samples before analysis of glucose equivalents by ion chromatography.

Destructive harvest

The plants were harvested with four replicates per genotype after 48 h of recovery. Fresh weight vegetative material including leaves and stems (FW-Veg) and fresh weight generative material including flowers and pods (FW-Gen) were immediately determined after cutting. After determining the fresh weights, the material was dried in an oven for 21 h at 80 °C. Dry weight vegetative (DW-Veg) and dry weight generative (DW-Gen) materials were measured afterwards.

Data analysis

Treatment and genotype differences were assessed by using the statistic program SPSS (SPSS Inc., Chicago, IL, USA). In SPSS, an analysis of variance (ANOVA, Duncan) was performed. Results are shown as means ± SD (standard deviation).

## 3. Results

The $F_v/F_m$ of all genotypes significantly decreased under cold stress (except for Acc#7 at SD-1) compared to control, on average by 5% in SD-3, but all genotypes did recover after cold stress (Figure 1A). Desi showed significantly the lowest $F_v/F_m$ in SD-2 and SD-3 during cold stress compared to the other genotypes (Figure 1A). Under heat stress, Acc#7 had significantly lower $F_v/F_m$ in SD-3 (74% reduction) compared to control, while the PSII efficiency of the other genotypes was maintained (Figure 1B). At recovery, $F_v/F_m$ of Acc#7 and Acc#2 was significantly lower, by 100% and 16%, respectively, than the control, and $F_v/F_m$ of Acc#7 was significantly the lowest compared to the other genotypes (Figure 1B).

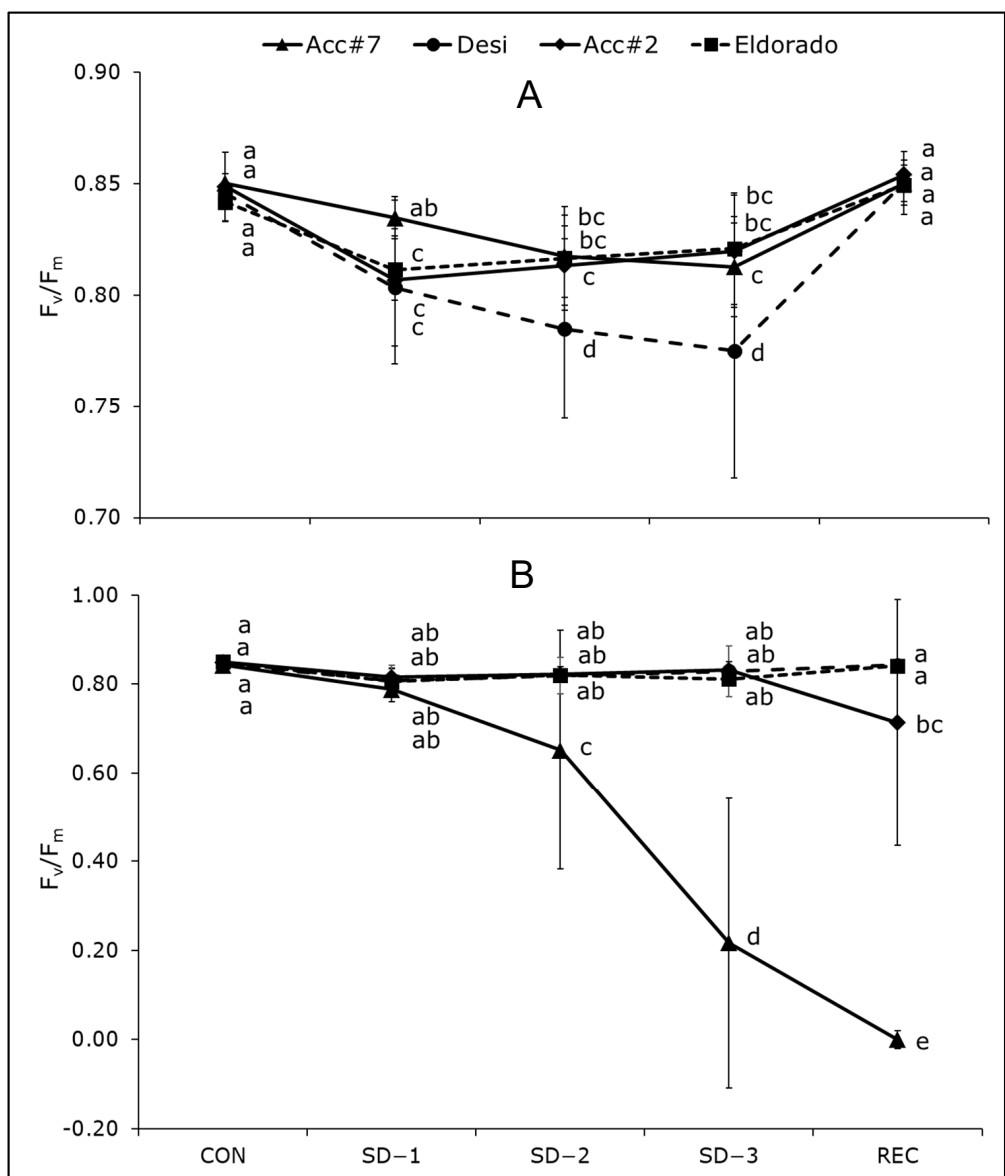

**Figure 1.** $F_v/F_m$ by mini PAM in the second fully expanded top leaves of four chickpea genotypes under the treatments for 3 days. "CON", 25/20 °C day/night T; "SD−1", after 24 h stress; "SD−2", after 48 h stress, "SD−3", after 72 h stress; "REC", after 48 h recovery time under control conditions. (**A** Cold-stress treatment: 8/4 °C day/night; (**B**) Heat-stress treatment: 38/33 °C day/night. The data represent average values ± SD (*n* = 8). Different small letters next to the point marks indicate significant differences (*p* < 0.05.).

The $P_N$ of all four genotypes significantly decreased under cold stress compared to that of control, on average by 74%; however, the $P_N$ of Desi decreased 94%, significantly the lowest level (Figure 2A). The $P_N$ of Acc#7 significantly decreased by 82% under heat stress in comparison with control, while the $P_N$ of Eldorado and Acc#2 significantly increased by 27% and 24%, respectively, overall (Figure 2A). The $C_i$ of Acc#2, Acc#7 and Eldorado was significantly by 32%, 14% and 35% under cold stress compared to control (Figure 2B). The $C_i$ of Desi and Acc#7 was significantly higher under cold stress than the $C_i$ of Eldorado and Acc#2 (Figure 2B). Under heat stress, the $C_i$ of Acc#7 significantly increased (11%) compared to control, while the $C_i$ of Eldorado and Acc#2 significantly decreased, by 17% and 12%, respectively (Figure 2B). The $C_i$ of Acc#7 was significantly the highest under heat stress compared to the other genotypes and treatments (Figure 2B). The $g_s$ of all four genotypes was significantly reduced under cold stress compared to control, on average by

95% (Figure 2C). Under heat stress, the $g_s$ of Acc#7 significantly decreased by 59% compared to control, and the $g_s$ of Desi was significantly higher compared to the other genotypes (Figure 2C). The E of all four genotypes was significantly reduced under cold stress, on average by 95% (Figure 2D). The E of Acc#7 was significantly decreased by 62% under heat stress, while the E of Acc#2 significantly increased by 41% (Figure 2D).

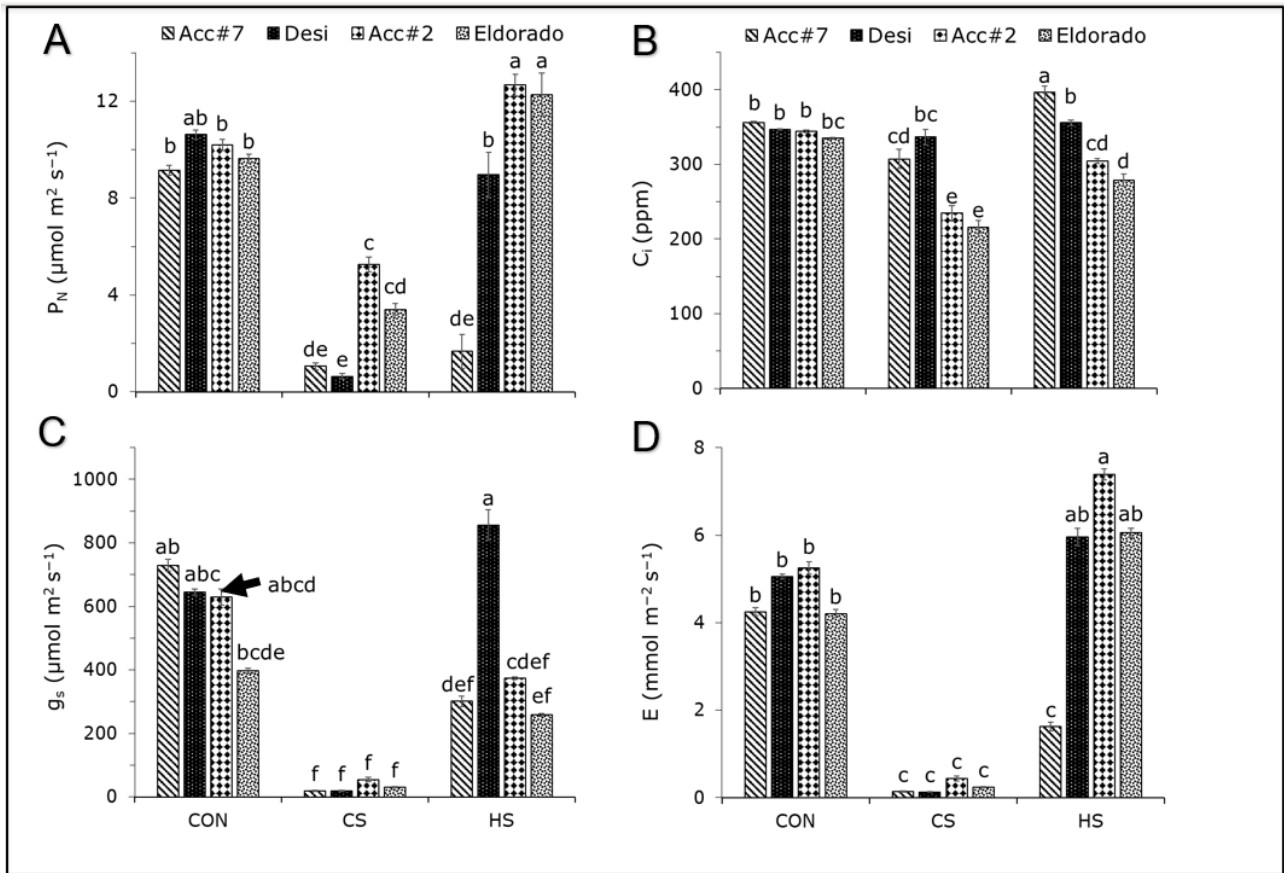

**Figure 2.** (**A**) Net photosynthetic rate ($P_N$), (**B**) intracellular $CO_2$ ($C_i$), (**C**) stomatal conductance ($g_s$) and (**D**) transpiration rate (E) in the second fully expanded top leaves of four chickpea genotypes under the treatments for 3 days. "CON", 25/20 °C day/night; "CS", 8/4 °C day/night; "HS", 38/33 °C day/night. The data represent average values ± SD (*n* = 3). Different small letters above the bars indicate significant differences (*p* < 0.05). The arrow is indicating that the letters abcd belong to Acc#2 under control conditions.

The Chl. a, Chl. b and Chl. a+b significantly decreased in the genotypes under both stress conditions, on average by 30%, 35% and 31%, respectively, under cold stress and by 42%, 30% and 39%, respectively, under heat stress, except for genotype Acc#2 under heat stress (Figure 3A–C). Under cold stress, Chl. a, Chl. b and Chl. a+b of Desi were significantly the lowest compared to the other genotypes, while those of Eldorado were significantly the highest (Figure 3A–C). Under heat stress, Chl. a, Chl. b and Chl. a+b of Acc#7 were significantly the lowest compared to the other genotypes, while those of Eldorado were significantly the highest (Figure 3A–C). The Chl. a:b ratio of Eldorado significantly increased under cold stress by 15%, while the Chl. a:b ratio of Acc#7 (45%) and Acc#2 (13%) was significantly decreased under heat stress (Figure 3D). The Chl. a:b ratio of Acc#7 was significantly lower under heat stress than the Chl. a:b ratio of the other three genotypes (Figure 3D). Furthermore, Chl. a:b was in general significantly lower under heat stress compared to cold stress (Figure 3D). The carotenoid content in Desi and Acc#7 was significantly reduced under both stress conditions, on average by 38% and 50%,

respectively (Figure 4). The carotenoid content in Desi was significantly the lowest under cold stress, while that of Acc#7 was significantly the lowest under heat stress (Figure 4).

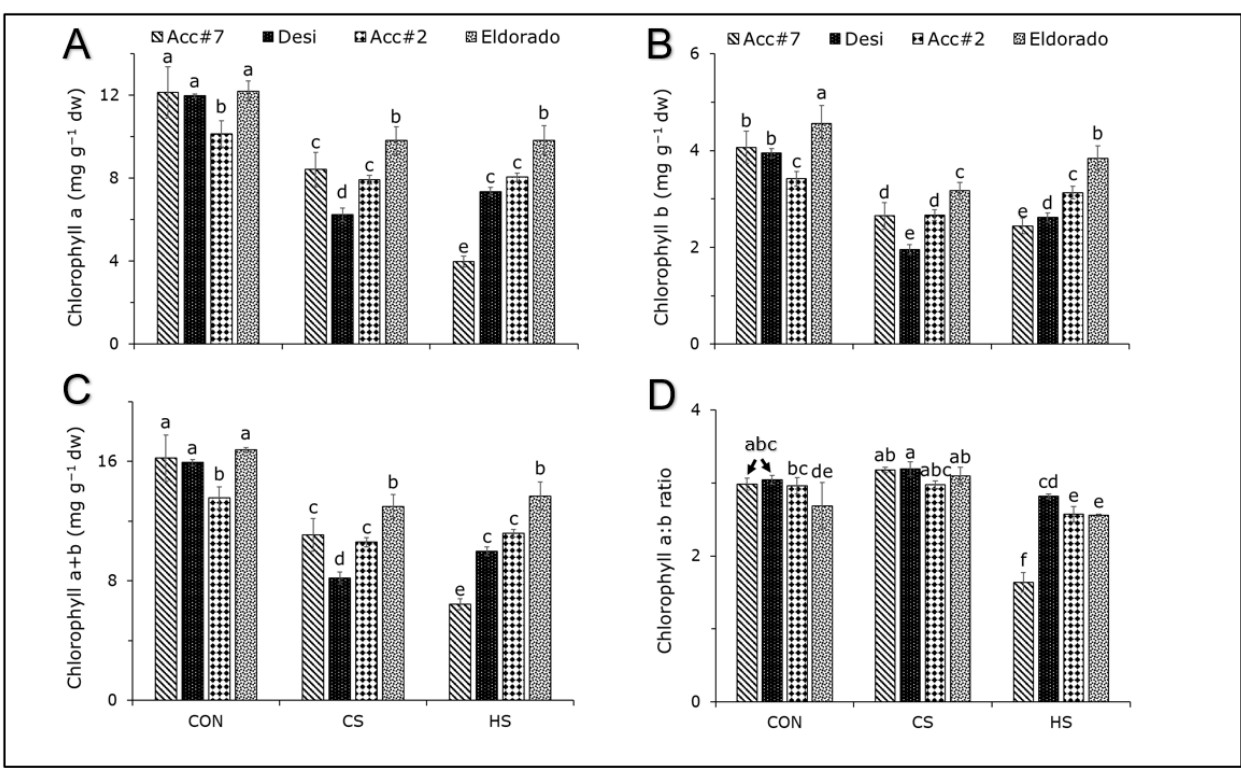

**Figure 3.** (**A**) Chlorophyll a (Chl. a), (**B**) chlorophyll b (Chl. b), (**C**) chlorophyll a+b (Chl. a+b) and (**D**) chlorophyll a:b ratio (Chl. a:b ratio) in the second fully expanded top leaves of four chickpea genotypes under the treatments for 3 days. "CON", 25/20 °C day/night; "CS", 8/4 °C day/night; "HS", 38/33 °C day/night. The data represent average values ± SD (*n* = 3). Different small letters above the bars indicate significant differences (*p* < 0.05). The arrows are indicating that the letters abc belongs to Acc#7 and Desi at control conditions.

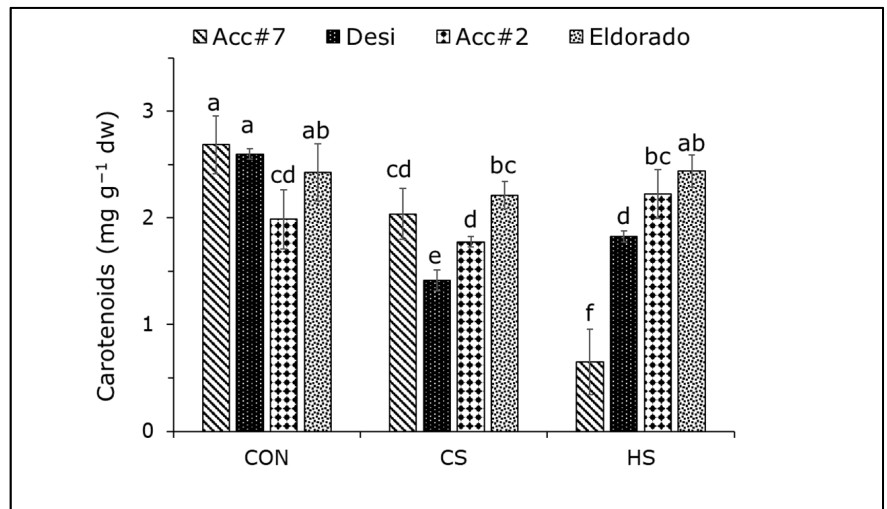

**Figure 4.** Carotenoids in the second fully expanded top leaves of four chickpea genotypes under the treatments for 3 days. "CON", 25/20 °C day/night; "CS", 8/4 °C day/night; "HS", 38/33 °C day/night. The data represent average values ± SD (*n* = 3). Different small letters above the bars indicate significant differences (*p* < 0.05).

Glucose, fructose, sucrose and starch of all genotypes significantly increased by 131%, 173%, 195% and 947%, respectively, under cold stress in comparison with control, except for glucose of Acc#2 (Figure 5). Sucrose and fructose of Desi were significantly the highest under cold stress compared to the other genotypes (Figure 5B,D). In contrast, glucose and fructose of Acc#7 and Acc#2 were significantly the lowest under cold stress (Figure 5A,B). Starch of Acc#2 had a significantly higher level under cold stress (Figure 5D). Glucose and fructose content in all genotypes significantly decreased by 77% and 76%, respectively, under heat stress, except for glucose in Acc#7 (Figure 5A,B).

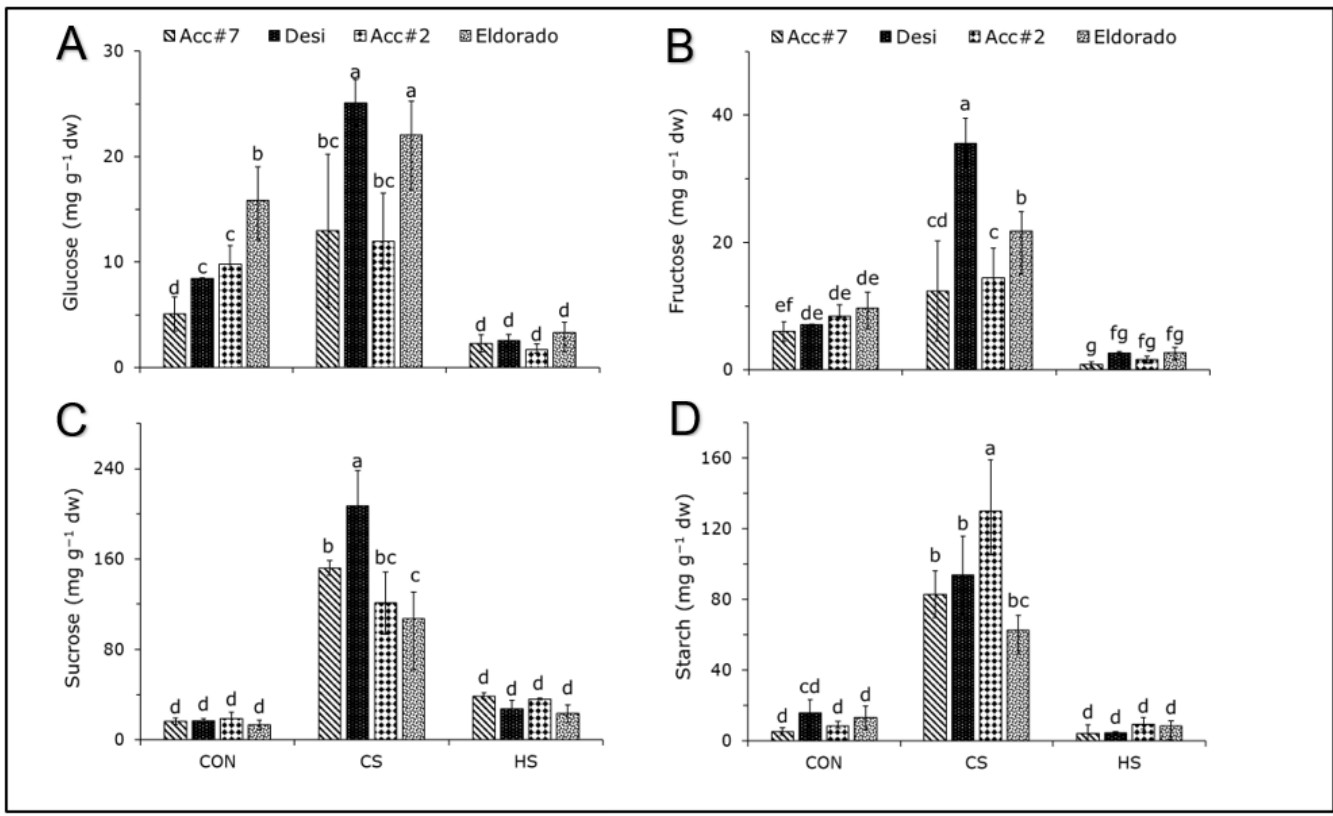

**Figure 5.** (**A**) Glucose, (**B**) fructose, (**C**) sucrose and (**D**) starch in the second fully expanded top leaves of four chickpea genotypes under the treatments for 3 days. "CON", 25/20 °C day/night; "CS", 8/4 °C day/night; "HS", 38/33 °C day/night. The data represent average values ± SD (*n* = 3). Different small letters above the bars indicate significant differences (*p* < 0.05).

The FW-Veg and DW-Veg of Eldorado were significantly the highest irrespective of treatments (Figure 6A,B). The FW-Veg of Acc#7 and Acc#2 was significantly reduced by 55% and 46%, respectively, under heat stress, where Acc#7 was the lowest compared to control and the other genotypes (Figure 6A). The unstressed levels of FW-Gen and DW-Gen of Acc#2 were the highest and significantly decreased by 84% and 78%, respectively, under heat stress (Figure 6C,D). The DW-Gen of Acc#2 was significantly reduced, by 33%, under cold stress compared to control (Figure 6D).

Under heat stress, the lower leaves of all genotypes turned yellow or even brown and fell off (Supplementary Figure S1). The Acc#7 was most sensitive to heat stress, as three out of four replicates died. Under cold stress, all genotypes maintained a normal appearance, and Eldorado even turned greener during the cold-stress treatment.

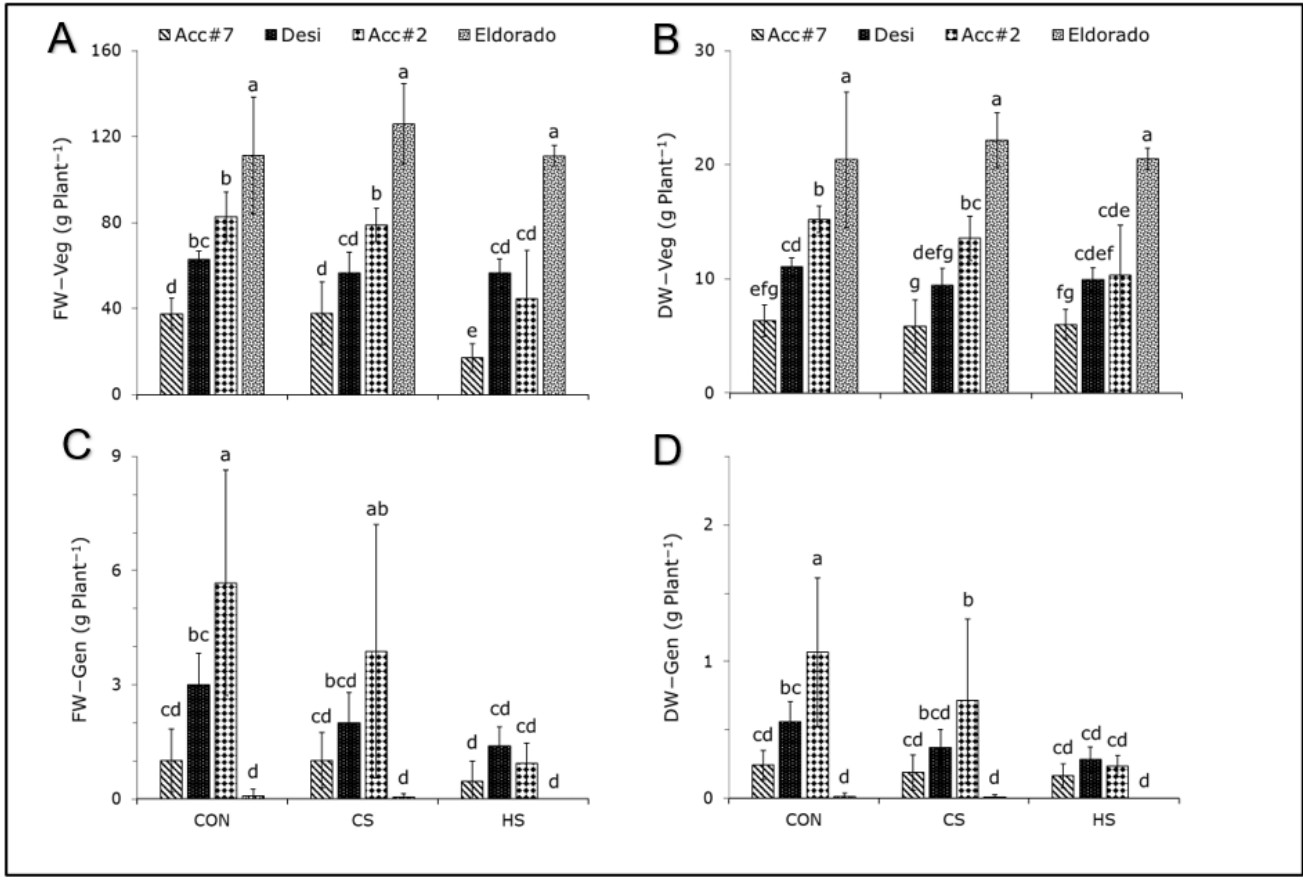

**Figure 6.** (**A**) Fresh weight vegetative (FW−Veg), (**B**) dry weight vegetative (DW−Veg), (**C**) fresh weight generative (FW−Gen) and (**D**) dry weight generative (DW−Gen) of four chickpea genotypes under the treatments for 3 days. "CON", 25/20 °C day/night; "CS", 8/4 °C day/night; "HS", 38/33 °C day/night. The data represent average values ± SD ($n = 4$). Different small letters above the bars indicate significant differences ($p < 0.05$).

## 4. Discussion

### 4.1. Effect of Cold Stress on Photosynthetic Apparatus and Carbohydrate Accumulation in Chickpeas

Exposure of plants to cold stress causes reduction and impairment of photosynthesis [5], and the reproductive phase is more sensitive to cold stress than the vegetative phase [3]. Low temperatures during flowering can lead to reduced pollination, flower shedding, pod abortion, poor germination, and in the worst case, to growth reduction, necrosis, and plant death [23]. Chlorophyll fluorescence provides the physiological status of temperature-stressed plant leaves, and decreases in $F_v/F_m$ can give insights into plant stress tolerance [24]. Cold stress led to decreased $F_v/F_m$ in faba bean [25] and chickpea seedlings [26]. Accordingly, cold stress significantly reduced $F_v/F_m$ in all tested chickpea genotypes, indicating that the efficiency of PSII photochemistry was adversely affected. Moreover, gas exchange parameters and the total chlorophyll content in leaves of all tested genotypes significantly decreased under cold stress, leading to reduced photosynthesis capacity. Chickpea has a strong indeterminate growth habit and great plasticity [27], which may serve as the mechanism of recovery in chickpeas [15]. It seems that all genotypes managed to recover fully after 3 days of cold-stress treatment, as there was no significant difference in $F_v/F_m$ between recovery and control, indicating a good recovery capacity.

Sugar and starch accumulations are common responses of plants to low temperatures, and by stabilizing cell membranes, they can contribute to cold tolerance in plants [3]. Increased carbohydrate accumulation was found in cold-acclimated 14-day-old chickpea

seedlings after 10 days of cold stress (4 °C), and sucrose accumulated to a greater extent, indicating a larger role in cold response [28]. Cold stress led to significantly increased sugar and starch concentrations in leaves within all tested chickpea genotypes, indicating their cold tolerance and ability to recover fast after cold stress. Moreover, sucrose accumulated most compared to the other carbohydrates, supporting its important role in cold response. Kaushal et al. (2013) [29] argued that the increased content in reducing sugars in chickpea leaves was related to an increased catabolism of starch and sucrose, which was also explained by Ruelland et al. (2009) [3] Here, the non-significant decrease in fresh vegetative and reproductive biomass of the four genotypes under short-term cold stress could be due to increased carbohydrate content in the leaves.

Apart from stress type, stress intensity, the occurrence of other stresses, and the phenology stage, the response and performance of chickpea to stress depends on the genotype [28]. There is genetic variation in chilling response, with some chickpea genotypes maintaining pod number even after they were exposed to 5 °C [12]. $F_v/F_m$, $P_N$, chlorophyll a+b and carotenoids were significantly lowest in Desi during cold stress compared to the other genotypes, indicating that Desi was a cold-sensitive chickpea candidate. Moreover, Acc#2, with the highest amount of generative biomass, showed a significant reduction in DW-Gen and $F_v/F_m$ after the cold-stress treatment, and it showed no significant difference in glucose concentration under cold stress compared to control, indicating its sensitivity to cold stress. In contrast, the late-flowering Eldorado, with the lowest generative biomass, had the highest chlorophyll a+b content and FW-Veg under cold stress compared to the other genotypes, indicating that it is a cold-tolerant candidate. The genotypic difference could be due to the different expression levels of key genes that are responsible for cold tolerance [30]. Clearly, further validation especially in the field for chickpea genotypes with high yields under cold stress is crucial to obtain cold-tolerant chickpeas and ensure their growth and production in areas with periods of cold weather.

### 4.2. Photosynthetic Heat Response and Its Strong Dependency on the Chickpea Genotype

By comparison, chickpea is classified as the most heat-sensitive legume, compared with others such as pigeon pea, groundnut, and soybean [31]. Chickpea productivity can be severely affected by heat stress, which will occur more frequently due to predicted future climatic change conditions, highlighting the urgent need to investigate enhanced heat tolerance in chickpeas [9]. Heat stress during the reproductive phase of legumes was generally linked with reduced or no pollination and abscission of flower buds, flowers, and pods, causing substantial yield loss [6].

Heat stress (>32/20 °C) significantly reduced the photochemical efficiency and $g_s$ in chickpeas during the reproductive phase, with larger effects on heat-sensitive genotypes [29]. Similarly, photochemical efficiency decreased with increasing temperature in chickpea genotypes, with greater inhibition in heat-sensitive chickpea genotypes [32]. Desi, Acc#2 and Eldorado increased or maintained their $F_v/F_m$ and $P_N$ levels under heat stress, since they could maintain E and $g_s$, indicating their tolerance to heat. The ability to maintain leaf gas exchange under heat stress has a direct relationship with heat tolerance [33]. Kaushal et al. (2013) [29] observed that high $g_s$ under heat stress improved water status in heat-tolerant chickpea genotypes. In contrast to the physiological performance of the other three genotypes, $F_v/F_m$ and $P_N$ of Acc#7 were significantly reduced under heat stress, accompanied by decreased $g_s$ and E compared with control, indicating that Acc#7 is a heat-sensitive candidate. Sudden heat-stress application (45 °C) let to defoliation, leaf drying and flower abortion in early-flowering and pod-setting chickpeas, resulting in low yields [34]. All tested chickpea genotypes significantly reduced the chlorophyll a+b contents in their leaves under heat stress, showing their sensitivity to the heat stress application.

Chickpea responses to heat stress were genotype-dependent, whereas heat stress in heat-sensitive genotypes decreased net photosynthetic rates ($P_N$), relative water content, maximum quantum efficiency of PSII ($F_v/F_m$) and chlorophyll content in lower bottom leavesAcc#7 and Acc#2 significantly reduced their fresh biomass under heat stress, indi-

cating their sensitivity to heat stress. Kumari et al. (2020) [9] suggest that "STAY-GREEN" traits are an important indicator for selecting heat-tolerant chickpea genotypes. Heat stress led to higher photosynthetic rates and starch, sucrose, and grain yield in heat-tolerant chickpeas compared to heat-sensitive genotypes [20] Heat-sensitive chickpea genotypes showed lower sucrose contents in their leaves, resulting in lower sucrose content in the pollen, reduced pollen function, impaired fertilization and poor pod set [33]. Although sucrose and starch contents did not change significantly, glucose and fructose contents significantly decreased in all chickpea genotypes (except for the glucose content in Acc#7) under heat stress, indicating their sensitivity to heat stress. Moreover, Acc#2 significantly reduced its generative biomass under heat stress treatment compared to control, indicating that it is a heat-sensitive chickpea candidate. Makonya et al. (2019) also classified Acc#2 as a heat-sensitive chickpea candidate.

Clearly, further validation in the field for chickpea genotypes with high yields under heat stress is crucial to obtain heat-tolerant chickpea varieties and ensure their growth and production in future climate change scenarios.

## 5. Conclusions

Cold and heat stress have a strong impact on the physiology of flowering chickpea, especially for photosynthesis, leading to limited generative and reproductive organ development. The short-term cold-stress application had an impact on the physiology of flowering chickpeas by decreasing their $F_v/F_m$, gas exchange parameters and leaf chlorophyll content, leading to limited photosynthetic activity. However, it seems that the high carbohydrate accumulations in their leaves in cold conditions helped them to maintain their flowers and pods. In contrast, carbohydrates did not accumulate in chickpeas under the heat-stress treatment, reducing the biomass and causing reproductive organ losses. Although heat stress reduced leaf chlorophyll content in chickpeas, $F_v/F_m$ and gas exchange were maintained. Genetic variation in response to temperature stress among the chickpea genotypes was observed. Desi and Acc#2 were characterized as cold-sensitive candidates, whereas Eldorado was a cold-tolerant candidate. Acc#7 and Acc#2 were identified as heat-sensitive candidates, while Desi and Eldorado were heat-tolerant candidates. Further research efforts are needed to gain a better understanding of the physiology of flowering chickpeas under cold and heat stress. Furthermore, validation of the heat and cold tolerance of chickpeas in the field is another necessity to ensure high yields under climate change in the future.

**Supplementary Materials:** The following supporting information can be downloaded at: https://www.mdpi.com/article/10.3390/agronomy12112755/s1. Figure S1. Plant morphology of four chickpea genotypes at the treatments for three days: Control (25/20 °C day/night), heat stress (38/33 °C day/night) and cold stress (8/4 °C day/night). Table S1. Actual climate conditions in control, cold stress and heat stress chamber under treatment and control con-ditions. The data represent average values ± SD of Temperature (T, °C), relative humidity (RH, %), and light inten-sity (Light, $\mu$mol m$^{-2}$ s$^{-1}$) at day and night period.

**Author Contributions:** Conceptualization, M.Z. and C.-O.O.; methodology, M.Z.; software, M.Z.; validation, M.Z., R.Z. and C.-O.O.; formal analysis, M.Z.; investigation, M.Z.; resources, M.Z.; data curation, M.Z.; writing—original draft preparation, M.Z.; writing—review and editing, M.Z and R.Z.; visualization, M.Z.; supervision, R.Z. and C.-O.O.; project administration, C.-O.O.; funding acquisition, R.Z. and C.-O.O. All authors have read and agreed to the published version of the manuscript.

**Funding:** The study was supported by Aarhus University Research Foundation (AUFF grant, no. 30379).

**Conflicts of Interest:** The authors declare no conflict of interest.

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
