# Peer review of "Physiological Responses of Chickpea Genotypes to Cold and Heat Stress in Flowering Stage"

_agronomy, doi:10.3390/agronomy12112755_

Round 1

Reviewer 1 Report

The manuscript study the physiological responses of four flowering chickpea genotypes to three days treatments under cold and heat stress. This is a further investigation after (Makonya et al., 2019) and (Zhou et al., 2020). The study is largely interesting and can be accepted after careful revison.

Questions:

1. Method part: The plants were moved  from control (25/20°C) into the chambers in  cold (9/4°C) and heat (38/33°C) conditions without stress acclimation.  Would these treatments be different from natural climate changes, which often has a transition last for several days?

2. Why did you assay only Fv/Fm for three days but not for other parameters, such as photosynthetic rate, leaf pigments and so on, which do not destruct too much plants?

3. In these parameters, which one is more important to assess the  sensitivity/tolerance of flowering and seed seting to cold/heat stresses. Why did you think Eldorado is tolerant but ACC#2 genotype is sensitive to cold? Based on the weight generative, Desi and ACC#2 maybe two better genotypes and Eldorado had worse performance in flowing and seed setting although good in vegetative grwoth.

4. There are some problems in the figures: Fig.1 the legend of acc#2 was masked by letter A.  What's the arrows indicated in fig.2 subfigure C and fig.3 D?

Reviewer 2 Report

This paper presents some useful findings on the effect of cold and heat stress on chickpea at flowering stage.

Overall, it is a well designed experiment but need some revisions, especially for the presentation quality.

1. Title should be: Physiological responses of chickpea genotypes to cold and heat stress at flower stage.

2. Introduction is too long. Please reduce it by 50%

3. Materials and Methods: Leaf pigments and carbohydrate measurement methods should be elaborated.

4. Discussion is too short. Please explain all parameters and relate among them. Cite recent references.

5. References are not formatted as per MDPI style. Please fix them.

Author Response

Point 1: Title should be: Physiological responses of chickpea genotypes to cold and heat stress at flower stage.

Respond 1: I have changed the title to Physiological responses of chickpea genotypes at flower stage to cold and heat stress. 

Point 2: Introduction is too long. Please reduce it by 50%. 

Respond 2: I have shortened the introduction. Please see in the attached file!

Point 3: Materials and Methods: Leaf pigments and carbohydrate measurement methods should be elaborated.

Respond 3: I have added some information on the leaf pigments and carbohydrate measurements. Please see in the attached document! 

Point 4: Discussion is too short. Please explain all parameters and relate among them. Cite recent references. 

Respond 4: I have worked on the discussion. Please see the in attached file!

Point 5: References are not formatted as per MDPI style. Please fix them.

Respond 5: I have downloaded the MDPI style and fixed the references. Please see in the attached document. 

Reviewer 3 Report

The experiment conducted by the authors broadens the knowledge on Physiological response of flowering chickpea genotypes to cold and heat stress’. This manuscript is not appropriate for publication in Agronomy I emphasize to reject the manuscript.

1. There is no novelty in the work presented.

2. These parameters which measured by authors is not enough to discuss their title.

Overall, this study is limited and only gives information on some photosynthetic parameters and doesn’t provide any concrete conclusion.

The manuscript gives a few results connected to the influence of stress. However it is far away to describe the mechanisms and the causal relationships.

Moreover, I wonder that the authors classified the chickpea genotypes as tolerant and sensitive to stressful conditions. I do not know why they are considered in this way since no references to previous studies showing its tolerance or sensitivity to stress are provided. Moreover, the measured parameters are not enough to classified the used chickpea genotypes.

Author Response

Point 1: There is no novelty in the work presented.

Response 1: In the last years there has been done various research about the effects of abiotic stresses on the physiology and yield of chickpeas. There exist various recent literature on single or combined heat stress on chickpeas, partly when they were at flower stage. However, there has been done limited research in the field of cold stress, especially when chickpeas were at flower stage. Therefore, our research covered two important temperature extremes – cold- and heat stress - and provides new insights on the physiological effects of those stresses on chickpeas at flower stage.

Looking at the physiological parameters, there have been found some similar reactions at both stresses, like reduced chlorophyll a+b contents. However, there have been found also controverse reactions: Fv/Fm and Pn reduced at cold stress while it stayed the same or even increased at heat stress; sugar accumulated in chickpea leaves at cold stress, but stayed the same/decreased at heat stress; etc. These reactions give important insights to understand better the chickpea mechanism to cold- and heat stress at flower stage.

Moreover, 4 chickpea genotypes from 3 different seedbanks got evaluated, which gives also interesting results looking at the genotype differences.

Point 2: These parameters which measured by authors is not enough to discuss their title. Overall, this study is limited and only gives information on some photosynthetic parameters and doesn’t provide any concrete conclusion. The manuscript gives a few results connected to the influence of stress. However it is far away to describe the mechanisms and the causal relationships. Moreover, I wonder that the authors classified the chickpea genotypes as tolerant and sensitive to stressful conditions. I do not know why they are considered in this way since no references to previous studies showing its tolerance or sensitivity to stress are provided. Moreover, the measured parameters are not enough to classified the used chickpea genotypes.

Response 2: We have worked on the manuscript and made some changes, please see in the attached file!

Reviewer 4 Report

The research topic presented in the reviewed manuscript entitled "Physiological response of flowering chickpea genotypes to cold and heat stress” by Mareen Zeitelhofer, Rong Zhou and Carl-Otto Ottosen is strictly connected to the physiological reactions of plants to the stress conditions. In the paper, the authors studied how cold and heat stress acted on the chickpea during the flowering phase of plant growth. There is no doubt that this stage of plant development plays the key role in processes connected with the pod and seed production. Both heat stress and cold stress can negatively affect crop growth and ultimately yield. As the object of the study, the authors chose a plant species involved in the production of food which is widely known all over the world. For the physiological experiments performed and presented in the paper, the authors chose four genotypes of Cicer arietinum which were selected from a greater number of genotypes.

          In the Introduction section, the authors underlined the influence of temperature, and especially its extreme variations as major environmental factor that adversely affects plant growth and development, and decided about agricultural production in the world. Next, the authors give some information about the stress actions and reactions of plants. The researches paid particular attention to the different mechanisms of various physiological processes which are affected by different stress factors and focused mainly on the chickpea plant. It is worth emphasizing that the data given by the authors concerned the Cicer arietinum species, and among other things, the authors provided its place its origin.

          The second section of the paper i.e. Material and Methods consists of information that is typically given for this section, and here, concerns the genotypes selected for these studies. Then, the authors describe conditions of plant growth and methods of stress treatment. Subsequently, they described the methods of performed measurements such as chlorophyl fluorescence, gas exchange and measurement of leaf pigments and carbohydrates. Finally, the authors wrote about how they analyzed the data obtained during experiments. The methods were selected and used adequately to reach the planned goals of the research.

The next section of the paper - 3. Results is the most extensive part in which the authors present the effects of performed experiments, and the results obtained are described in detail and shown mainly in the form of diagrams and collected in 6 figures.

The section “Discussion” is short, but briefly presents the analyzed results in comparison to published studies. The conclusions in the section 5. of the paper summarized the main data obtained by authors in the proper way.   

The experiments performed and presented in this paper are properly carried out. They are also discussed in the proper way and sufficiently detailed. Generally, I am convinced that this paper is very interesting and suitable for publishing in the “Agronomy”.

Author Response

Thank you for sharing your comments. I am happy to hear that you are convinced to include this research in Agronomy. 

Round 2

Reviewer 3 Report

The experiment conducted by the authors is not appropriate for publication in Agronomy I emphasize to reject the manuscript.

In Conclusions section authors said that Genetic variation in response to temperature stress among the chickpea genotypes was observed. Desi and Acc#2 were characterized as cold-sensitive candidates genotypes whereas and Eldorado as a cold-tolerant candidate. genotype. Acc#7  and Acc#2 were identified as heat-sensitive candidates and genotypes and Desi and Eldorado as heat-tolerant candidates.

Indeed, authors did not measure any genetic parameters to conclude their conclusion. This study is limited and only gives information on some photosynthetic parameters.

Furthermore, I wonder that the authors classified the chickpea genotypes as tolerant and sensitive to stressful conditions. I do not know why they are considered in this way since no references to previous studies showing its tolerance or sensitivity to stress are provided. Moreover, the measured parameters are not enough to classified the used chickpea genotypes.
